# Divine Morality or Divine Love? On Sterba's New Logical Problem of Evil

Jonathan C. Rutledge 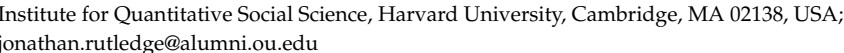

Institute for Quantitative Social Science, Harvard University, Cambridge, MA 02138, USA; jonathan.rutledge@alumni.ou.edu

**Abstract:** In his recent version of the logical problem of evil, James Sterba articulates several moral principles that, on the assumption that God is morally perfect, seem to entail God's non-existence. Such moral principles, however, only apply to God on the assumption that he is a moral agent. I first argue against this assumption by appealing to recent work by Mark Murphy before, secondly, suggesting an alternative way to frame Sterba's argument in terms of divine love. One can distinguish God's motivation to promote creaturely welfare on the basis of love from a motivation grounded in morality, and I claim that doing so results in a stronger form of the logical argument.

**Keywords:** problem of evil; rationality; value theory; divine agency

In *Is a Good God Logically Possible?*, James P. Sterba argues that commitment to certain moral principles—i.e., principles which follow from a suitably qualified version of the Apostle Paul's emphatic thesis to *never do evil that good may come* (Rom. 3:28)—rules out God's existence (Sterba 2019). Specifically, Sterba claims that once we account *not only* for the mere existence of evil *but also* for the particular kinds and distribution of horrendous evils that actually characterize our world, a sound logical argument against God's existence (roughly in the vein of the classic Plantinga–Mackie debate) can be constructed out of those moral principles Sterba identifies—see (Mackie 1955) and (Plantinga 1974, chp. 9).[1]

The aim of this essay is to present an opportunity for further discussion concerning the question of why theists should think that moral principles, whatever their content, apply to God. Thinking that such principles *do* apply to God requires that God is a moral agent in some sense, and given that such a claim has been a called into question by several serious philosophers and theologians, the idea that God is a moral agent is at least not analytically true.[2] In other words, there is room for debate here.

In chapter 6, Sterba considers one version of this objection to his argument—due to Brian Davies (2006)—that, contrary to the inclinations of many *contemporary* philosophers of religion, God's moral perfection does not follow from his perfect goodness. In what follows, I take a page out of some recent work by Mark C. Murphy in arguing that we have strong reason to deny that God is morally perfect (Murphy 2017, 2021). Even so, I think Sterba's argument can be repaired by appealing to a God of *love*, and I suggest how that repair might proceed (although I do not here suggest that the revised argument ultimately succeeds, for other objections to Sterba's argument, which I set aside for the purposes of this paper, are relevant in evaluating the revised version of his argument).

I begin by explaining Sterba's argument with a bit more precision before turning—in Section 2—to an explanation of how denying that moral norms apply to God undermines his argument. Then, in Section 3, I present arguments for why God's *absolute* perfection might be seen to preclude *moral* perfection, and close (Section 4) with an alternative version of Sterba's logical argument (henceforth, 'SLA') articulated in terms of divine love.[3]

## 1. Sterba's Logical Argument (SLA)

Sterba begins his argument with a discussion of the Pauline Principle to "never intentionally do moral evil that greater good may come of it" (8fn5). Immediately, he notes

that such a principle cannot hold universally since there are obvious scenarios in which it fails to hold—e.g., whenever one lies in order to protect the life of a friend. Nevertheless, Sterba takes there to be "exceptionless minimal components of the Pauline Principle" that *do* hold for all rational moral agents (including God), which he labels Moral Evil Prevention Requirements. They are:

(I)     Prevent, rather than permit, significant and especially horrendous evil consequences of immoral actions without violating anyone's rights (a good to which we have a right), as needed, when that can easily be done.

(II)     Do not permit significant and especially horrendous evil consequences of immoral actions simply to provide other rational beings with goods they would morally prefer not to have.

(III)     Do not permit, rather than prevent, significant and especially horrendous evil consequences of immoral actions on would-be victims (which would violate their rights) in order to provide them with goods to which they do not have a right, when there are countless morally unobjectionable ways of providing those goods.[4]

The rationale behind each of these principles is fairly straightforward. They all concern an agent who is considering whether or not to *intervene* to prevent the consequences of some evil; i.e., consequences they reasonably expect to follow without their intervention.[5] Whenever intervention (or lack thereof) would fail to satisfy any of I-III, then the agent under question must, on pain of acting immorally, refrain from intervening (or not) in accordance with the deliverances of the principles. Moreover, given God's omniscience, God would always be aware of how these principles bear on his deliberations for action. From this we get (my restatement of) Sterba's new form of the logical argument from evil (SLA), which I state in the form of an inconsistent set of propositions:

1.     An omnipotent and morally perfect being exists.
2.     An omnipotent being could always act in accordance with Moral Evil Prevention Requirements I, II, and III.
3.     A morally perfect being always acts in accordance with Moral Evil Prevention Requirements I, II, and III if it is possible to do so and that being is omnipotent.
4.     If a being always acts in accordance with Moral Evil Prevention Requirements I, II, and III and is omnipotent, then it is not the case that there exist significant and especially horrendous evil consequences of immoral actions that easily could have been prevented by an omnipotent being without violating Moral Evil Prevention Requirements I, II, or III.
5.     There exist significant and especially horrendous evil consequences of immoral actions that easily could have been prevented by an omnipotent being without violating Moral Evil Prevention Requirements I, II, or III.[6]

The argument to the inconsistency of (1)–(5) clearly holds.[7] And, moreover, it is structurally very similar to Mackie's own statement of the logical problem of evil.[8] What makes Sterba's version significantly different, however, is the existential claim found in premise (5): namely, that there exist not only evils—that was Mackie's rather minimal suggestion—but also evils that conflict with Sterba's moral requirements. In other words, Sterba provides a more demanding existential premise—i.e., more demanding in the sense that its truth requires us to affirm more about the world—that allows for the possibility that God's existence is compatible with *some* evil; it is just that God's existence is not compatible with the particular evils with which Sterba is concerned (i.e., the evils we witness in the world that violate Moral Evil Prevention Requirements I–III). Given that (1)–(5) constitute an inconsistent set, no one can consistently affirm all five claims. Sterba, of course, argues that we should deny premise (1), which amounts to a denial of God's existence on the assumption that God is a morally perfect and omnipotent being.

## 2. The Centrality of Moral Norms for SLA

Something that is immediately apparent after stating SLA in this way is its reliance on moral principles. For if one were to leave off these principles, then one would be leaving off those things which most distinguish SLA from Mackie's logical argument.

One thing worth noticing, however, is that Sterba's moral principles simply highlight something which is already built into Mackie's original argument: namely, that the being whose existence is being ruled out is said to be *morally perfect* (cf. claim (1) of the inconsistent set above). However, moral perfection presumes that we are talking about a *moral agent*—i.e., someone to whom the norms of morality apply—if it is to be informative at all. Suppose, for instance, that your friend tells you that their dog, Dante, is morally perfect. Confused, you press them: "What do you mean when you say Dante is morally perfect?" They respond, "Well, Dante has never done anything immoral, so surely he can count as morally perfect in some sense." You should not be convinced. Ascribing moral perfection to something only makes sense if it is the sort of being that *responds* (or can/should respond) to moral reasons. The fact that Dante is a dog along with the fact that dogs neither can nor should respond to moral reasons precludes Dante from being morally perfect in virtue of the fact that Dante is not a moral agent.[9]

Fortunately, unlike Dante, God is fully capable of comprehending the sorts of moral principles invoked by SLA and, in virtue of this, one might think that God is a moral agent. However, moral agency includes a normative component as well: namely, that when there is a moral reason to do something, then, in the absence of contrary reasons, a moral agent is *required* to act on it, on pain of irrationality. In other words, for someone to be a moral agent, the moral considerations under question (e.g., beneficence, dignity of persons, etc.) must provide them with *requiring reasons* for action. Furthermore, in the case of SLA, what puts pressure on the theist is that Sterba takes pains to explain how God, in particular, would never have reasons to act against Moral Evil Prevention Requirements I-III. That is, God would always have *decisive* reason to follow such norms.

Let us take a moment, however, to explicate the above distinction—i.e., that between requiring and (merely) justifying reasons for action—with a bit more care in relation to SLA. First, as just mentioned, a requiring reason is a reason that *constrains* action; that is, if an agent fails to act on such a reason, in the absence of contrary reasons, then that agent acts irrationally. Of course, the agent might have reasons to the contrary, and if so, then their choice to act against such a requiring reason need not be irrational. However, if that agent fails to have contrary reasons, then, insofar as acting rationally is interchangeable with acting as reason dictates, the agent is irrational.

Secondly, *what sorts of reasons might make it rational to not act in accordance with requiring reasons?* One answer to this question is other requiring reasons. For instance, suppose a doctor has a requiring reason to act for the benefit of her patients. When faced with a patient in need of immediate treatment—e.g., an epinephrine shot to counteract an allergy—the doctor is bound by requiring moral reasons to implement the relevant treatment. However, suppose the patient has previously, in full awareness of the life-threatening potential of his allergy, declared that under no circumstances, even lifesaving ones, does he consent to receiving a shot of epinephrine. In that case, the doctor will have a requiring reason to *not* implement the treatment, i.e., a reason grounded in respect for her patient's autonomy.

A second answer to our question—i.e., *why might someone reasonably not act in accordance with one's requiring reasons*—would appeal to (merely) justifying reasons for action, where a (merely) justifying reason[10] for action presents an agent with rational opportunities for action without constraining the agent's action in any sense. To put it a bit more technically, a justifying reason for action makes an action in accordance with it rational, but should an agent fail to act on such a reason, even in the absence of reasons to the contrary, she is not thereby rendered irrational. Rather, she has simply chosen not to pursue something she has some reason to pursue.

Examples of justifying reasons like this are ubiquitous. There is value in pursuing a Ph.D. in philosophy but there is also value in pursuing a law degree. Faced with both

options, an agent may choose to pursue one of them without pursuing the other. Suppose I choose to pursue a law degree. Does this imply that my decision to not choose the Ph.D. in philosophy was irrational? Not at all, for the decision with which I was faced was a decision between reasonable *opportunities*.[11]

Can such justifying reasons make it rational to act against one's requiring reasons? It can, and this is easy to see especially when the strength of a requiring reason is low. Suppose that I have promised to meet a group of friends for lunch. Keeping promises grounds requiring reasons of morality on my view, and so, I have a requiring reason to meet my friends for lunch.[12] However, on the way to lunch, I walk next to the oddest thing: a phone booth where passersby are invited at random to step into the phone booth before it is filled with gusts of wind and a mix of real and counterfeit Bruce Springsteen concert tickets. Obviously, the goal for each participant is to get at least one genuine Bruce Springsteen ticket, and, to my great delight, I am invited to participate. Even so, I hit a snag, for I must wait half an hour before being granted entry, i.e., something which would preclude my making it in time for lunch. Suppose I choose to wait. Am I acting irrationally? I might be acting *immorally*, for, after all, I am acting against a requiring reason of morality without having a moral reason to the contrary. Even so, my decision to try to obtain a Bruce Springsteen ticket is *intelligible*, even if one disagrees with it. Why? Well, my friends understand my Bruce Springsteen obsession, and when I tell them about it, they won't be hurt by my moral indiscretion. They know me well enough to be amused instead of hurt by such a decision, and supposing that I am aware of this, the decision becomes all the more understandable. That is, this is the sort of decision we can imagine a rational actor making without the decision being *required* by rationality (even if it were required by morality). The reason on which I act is a practical justifying reason for action, and it provides a rational opportunity for acting contrary to my moral requiring reason.

Third, consider the notion of a *decisive* reason for action. Whenever one has a requiring reason to act but there are no reasons to the contrary, then that requiring reason is also *decisive*. That is, that reason alone entails what a perfectly rational actor would do (for acting against it can be rational only in accordance with reasons to the contrary; but you have no reasons to the contrary on the supposition). Thus, if God is perfectly rational, then anytime God has a decisive reason to φ, God φ's.[13]

Given all of this, let us return to SLA. In that argument, Sterba assumes that God is responsive to moral considerations—that is, moral considerations give God justifying reasons, at least, for acting. He does not stop there though, for, on Sterba's account, such reasons are also both (i) *requiring* for God and (ii) *decisive* when considering situations that might conflict with Moral Evil Prevention Requirements I-III. Notice, then, that if there were grounds to deny that moral reasons were rationally requiring for God as a perfect being, then SLA would be undermined. Are there any such grounds?

## 3. Divine Perfection Does Not Entail Moral Perfection

To ascertain whether or not there are grounds for thinking of God as possessing *requiring* reasons of morality, it is worth beginning with a lengthy quote from Sterba in which he responds to Brian Davies. On the view of Davies that Sterba is considering, were God to command human agents to torture children, that would involve God in a contradiction since God has built *not* torturing children into human nature itself. That is, were God to command us (humans) to torture children, according to Davies, then God would be *explicitly* commanding us to act in contradiction with those commands that are *implicit* within the human nature God has given us. Sterba responds to this idea as follows,

> [I]f God cannot command us to do anything that goes against the law of reason that he embedded in our hearts because that would involve God in a contradiction, then, it would also seem that God could not act against that same law of reason that he embedded in our hearts because that too would involve God in a contradiction. Thus, God, for example, like us, would be required not to torture

children. However, Davies rejects this result on the grounds that it would make God a moral agent with moral requirements or obligations like ourselves.

How can Davies hold this? If it would be contradictory for God to implant a law in our nature that applies to all rational agents and then command that we act against it, why would it not also be contradictory for God to implant a law in our nature that applies to all rational agents and command us not to act against it, while at the same time regularly acting against the law himself in his dealings with us? (Sterba 2021, p. 116)

After restating Davies's position, Sterba reacts in befuddlement ("How can Davies hold this?"). Then, in articulating *why* he is befuddled, Sterba, in representing the commitments of Davies's view, claims that God has implanted "a law in our nature that applies to all rational agents". Although it is easy to miss, this appears to be a misrepresentation of Davies's view (or if it is not a misrepresentation, then Davies should change his view). For even if God has put a law in our nature that applies to all rational *human* agents, it does not follow that it applies to all rational agents. In other words, Sterba is assuming a sort of universal applicability that comes with such moral considerations. Specifically, Sterba assumes that the sorts of welfare values underlying the moral norms in question, first, give all rational agents *reasons* to act, and second, that such reasons would all be the same *kinds* of reasons.

Welfare values are relative to specific agents, and the presence of such values alone does not ground requiring reasons for every agent. For instance, *having sufficient nutrition* is a valuable state for my dog, and I tend to think, in virtue of the relationship I bear to my dog, that this value gives me a requiring reason to feed her (a requiring reason not shared widely if at all). Indeed, were someone to try and feed my dog without my permission, I might be reasonably upset. It is *my* responsibility to take care of her, and, on the assumption that I am fulfilling that responsibility, other agents are precluded from doing so. In other words, other agents seem to have requiring reasons *not* to feed my dog (despite the fact that doing so is to aim at a valuable state of affairs for my dog).

Even so, the sorts of scenarios involving welfare values for human agents imagined by Sterba carefully stipulate the conditions in question such that were such welfare values to provide humans with moral reasons for action, then they plausibly provide requiring reasons for all humans (i.e., irrespective of one's special relationships to those in danger of suffering). Thus, if they apply universally to human agents, then it is a natural thought that they also apply, and provide requiring reasons for action to, divine agents.

Is this natural thought correct, though? To see that it need not be, begin by noticing what my dog feeding example illustrates: namely, that there are circumstances in which the fact that (1) *X is fundamentally good (bad) for A* does not entail that (2) *X is a reason for all agents to promote (prevent) X*.[14] That is to say, the inference from (1) to (2) is not definitional or a matter of the nature of value (goodness/badness) itself. Rather, one must appeal to *other factors* to explain why the value under question applies to a given agent in such a way that they have a requiring reason for action.

One might try to make the move from (1) to (2) more manageable by qualifying (2) in various ways. For instance, perhaps rather than taking values to ground reasons for *all agents* to act, we might identify a particular *class* of agents as the relevant one. Aristotelians, for example, explain what it is to be a good human by appealing to action in accordance with *human* virtue. Accordingly, human agents all have requiring reasons to promote certain human goods in virtue of being human (rather than in virtue of being rational agents of some kind or another).[15] Since God is not human, it seems God does not have requiring reason to promote human-specific values on such an Aristotelian account. For yet another case of this, Humeans ground the reasons-conferring power of value in shared human moral sentiments such that, for any agent that possesses such sentiments, that agent has requiring reason to promote that value. Even on this account, however, such sentiments do not appear rational in any strong sense. That is, there is no reason to think that rational agency stands or falls on the possession of such sentiments, and if that is correct, then one

will be hard pressed to demonstrate that such Humeanism can show that welfare values are sufficient to ground requiring reasons of morality for a non-human, divine rational agent. Shared sentiments might ground universal *human* applicability, but they, by no means, ground universal *rational agent* applicability.[16]

Suppose we instead try to bridge the inference from (1) to (2) in Kantian fashion; that is, suppose we try to show that failure to promote welfare values for humans involves some sort of contradiction in reason itself. Perhaps such a project would indeed demonstrate that promoting welfare-oriented goods—like those involved in Sterba's Moral Evil Prevention Requirements I–III—is required not only of humans but also any rational agent. If successful, then, such a project would establish that God does indeed have requiring, even decisive reason to govern the world in a way that is consistent with Sterba's Moral Evil Prevention Requirements I-III. However, Sterba provides no suggestion about how that project might proceed; rather, he simply assumes that it can be accomplished without argument. As a result, we are left without sufficient reason to understand God as a moral agent in the sense required by SLA.[17]

### 4. Love's Non-Comparative Nature

So, SLA does not succeed in its current form to demonstrate the impossibility of God's existence. Even so, were Sterba able to secure the connection between God and God's having requiring reasons to promote human welfare in some way other than assuming that God is a moral agent, a renewed version of his argument might be successful. What other way might be available to Sterba?

To see what might succeed here, it is worth remembering that the logical problem of evil is supposed to be a problem for *theists*. Contrary to this, one (regrettably) sometimes hears theists respond to the problem of evil by saying something of the form, "Well, atheists don't believe in evil anyway, so there's no problem on their worldview!" Of course, plenty of atheists (e.g., Sterba himself) believe in the existence of moral evil, and they have plausibly adequate grounds for doing so. However, what an atheist believes about morality is irrelevant to the problem of evil. What matters is what *theists* believe, and this presents Sterba with an alternative route to a successful logical argument.

Even if a theist denies that God is a moral agent, most of them will affirm that God *loves* humanity.[18] Among other things, if *A* loves *B*, then that entails that *A* desires *B*'s ultimate good. But not only does love entail desiring the beloved's good; love is plausibly a noncomparative state that grounds reasons for the lover to act independent of the comparative value of the beloved to other creatures. Allow me to say a bit more about this.

I love my spouse tremendously. My love for her might manifest in my cooking a hot breakfast each morning for her, expressing my gratitude and delight in seeing her each day, in encouraging her to pursue her many talents in various ways, etc. However, and this is crucial, my love for my spouse has nothing to do with how much better she is than other people or how much more suitable as a spouse she is for me. Indeed, were someone to ask me why I love her, if I responded by offering a series of *comparisons* for why it is that she is better than all other potential spouses, they could rightly question whether I truly loved her. Love involves a response to the beloved that is noncomparative, such that genuine love gives one reasons to promote the welfare of the beloved, no matter how they compare to other humans, creatures, agents, etc. Questions of the objective value they hold simply have no primary relevance in determining whether one seeks their good. One values them in loving them and that gives one reason to act to promote their good.[19]

Accordingly, I submit that if God loves someone, then God has requiring reasons to promote their welfare in the way that SLA suggests. Thus, if Sterba were to replace 'morally perfect' with 'perfectly loving' throughout all the premises of my earlier reformulation of his argument, the question of the argument's success against God's existence would no longer turn on a concept of God rejected by many theists but would instead hinge on the evaluation of the suitably changed existential premise—i.e., premise (5).[20]

Allow me, however, to briefly respond to one further worry inspired by Mark Murphy. In *God's Own Ethics*, Murphy contends that whatever reasons of love God has to promote creaturely well-being really amount to nothing more than moral reasons to promote creaturely well-being. Clearly enough, if Murphy is correct in this claim, then replacing 'morally perfect' with 'perfectly loving' would make no substantive difference to the success of Sterba's argument, and my suggested fix would amount to dust.[21]

My response to Murphy, however, goes as follows: the reasons we have that are moral are, as I understand them, grounded in the objective value of other agents. That is, we have requiring moral reasons to promote the welfare of others in virtue of the dignity they possess and the fact that our own dignity is not saliently different from theirs. This is a Kantian way of thinking about the application conditions of moral reasons to all human agents.

Very plausibly, God's own objective dignity and value surpasses our own. Well... perhaps that puts things too delicately. Perhaps I should instead say that God's value is so great that, by comparison, our objective value bears *no significance* on God's deliberations whatsoever. That is, our objective value gives God no more (in fact, less) reason to promote our welfare than the objective value of an ant gives humans reason to promote its welfare. This is part of the reason why treating God as a moral agent runs into difficulties. Notice here, however, that the entire argument for Murphy hinges on the *objective* value gap that holds between God and human agents, leaving the subjective value conferred on human agents by God's love of them to the side.

When we do attend to God's love of human agents (and the reason-conferring nature of love), things are different. The reasons of love God possesses are not responses to our objective value (at least, not fundamentally so); rather, they are grounded in God's response to *us*. That is, they arise out of a divine love for who we are, no matter how we compare to other agents or creatures.

What this means is that God's decision to love human agents gives God reason to act so as to satisfy the welfare-based moral principles articulated by Sterba. Moreover, to try and ground the reasons of love in something other than God's choice to love humans—e.g., the objective value of human agents—is to treat them, instead, as moral reasons that apply to all rational agents (and we have already seen that grounding moral constraints that apply to a divine being requires further argumentation than Sterba has yet provided).

In virtue of this, Murphy's claimed reduction of reasons of love to reasons of morality, at least with respect to what requiring reasons God would have to promote our good, fails. Thus, if a theist takes God to be perfectly loving, then they can simultaneously deny that God is a moral agent while still affirming that Sterba's Moral Evil Prevention Requirements I-III articulate conditions for action that would give God decisive reasons to act in various ways. If this is right, then Sterba's argument, reconstructed in terms of divine love, presents a more powerful version of the logical argument (though I hereby refrain from issuing any verdicts on the ultimate success of the revised argument).

## 5. Conclusions

In this essay, I have argued that, contrary to his contention in chapter 6 of *Is a Good God Logically Possible?*, Sterba has not provided reason to think that God is a moral agent. This, of course, is a problem for SLA since it is predicated on the assumption that God's moral agency clearly follows from God's perfection. Nevertheless, I have suggested that once one appreciates the difference between the nature of love and reasons of morality, a stronger version of Sterba's logical argument can be developed which reestablishes the connection between the God of theism and the expectations Sterba articulates regarding divine action. Thus, even if God is not a moral agent, there remains a serious logical problem of evil facing theism.

**Funding:** This research received no external funding.

**Conflicts of Interest:** The author declares no conflict of interest.

## Notes

1.   Sterba divides all goods into (i) first-order and second-order goods—where second-order goods are those goods which logically require the existence of a prior evil (e.g., comforting a survivor of abuse)—and (ii) goods to which we have a right and goods to which we do not have a right. It is in exploring God's relationship to (or permission of) various combinations of these types of goods and evils that Sterba's argument (or arguments) rule out God's existence.

2.   I take the philosophers and theologians who deny that God is, necessarily, a moral agent to understand terms such as 'God' or 'divine' sufficiently well that if God's moral agency were a mere matter of the definition of 'divinity', then they would not call God's moral agency into question. The fact that they *do* question it, then, strongly implies (even if it fails to guarantee) that the idea that God is a moral agent is not analytically true.

3.   This final section turns on the notion that love is noncomparative, and so, it can plausibly provide the needed universal applicability for welfare-based reasons to motivate God to act in the way Sterba needs.

4.   These requirements are explicitly formulated first in (Sterba 2021, pp. 126–28), although this version of the principles is taken from 184.

5.   Of course, in God's case, God does not only reasonably expect the consequences to follow. He, on account of his omniscience, *knows* that they would follow without his intervention.

6.   Sterba (2019, pp. 181–94) helpfully offers several restatements of his various arguments and sub-arguments. The above is my own restatement aiming to display the formal similarity between Sterba's argument and Mackie's older logical argument. Note, also, that given that (1)–(5) are in fact inconsistent, rationality does not *require* giving up claim (1). Rationality, rather, requires giving up at least one of claims (1)–(5) if one wants to avoid inconsistency.

7.   See (Weber 2019) for a response to anyone wishing to push a paraconsistent approach to arguments from evil.

8.   Mackie, of course, puts the argument in terms of an omnipotent and "wholly good" being, but it seems fairly clear that he has a *morally perfect agent* in mind (see "Evil and Omnipotence").

9.   Perhaps one should not assume that Dante is an agent in the example. I am not inclined to this, but if you are, fine. In that case, my conclusion can be reached even more easily, for if Dante isn't an agent, then Dante is certainly not a moral agent.

10.   From here on out, if I say 'justifying reason', I mean a (merely) justifying reason.

11.   (Murphy 2017, p. 59) uses this terminology of 'constraints' and 'opportunities'.

12.   One reviewer notes the complexity of the moral nature of promise keeping and suggests I flag it here: if you doubt that making a promise grounds at least a *prima facie* requiring reason of morality for someone to act, then substitute your preferred ground for moral reasons. If someone thinks there are no requiring reasons of morality, then that disagreement would completely undermine the success of Sterba's argument. For if no one has requiring reasons of morality, then God clearly does not either. Given that this example is just an illustration of the requiring/justifying reasons distinction, one's choice of moral requiring reasons makes little difference.

13.   For these distinctions in types of reasons, see (Gert 2004). See also (Murphy 2017, 2021) for arguments employing the distinction to the effect that God is not a moral agent.

14.   The portions that are italicized in my text here come directly from (Murphy 2017, p. 49), though Murphy does not italicize them. His reflections in chapter 3 of that book guide the vast majority of my discussion in this section.

15.   Technically, we run into a difficulty on Aristotelianism here since one might think that one ought to merely promote the human good *for oneself* in virtue of being human without also being constrained to promote that good for others as well. I set aside this issue for the sake of argument. See (Wolterstorff 2010) for a discussion of this issue in some strands of *eudaimonism*.

16.   See (Murphy 2017, pp. 49–58) for a more thorough explanation for the difficulties faced by various ethical theories in grounding the universality of moral norms for all rational agents. My comments here on Humeanism and Aristotelianism essentially follow Murphy's own argument there. I am streamlining that discussion due to the space limitations here.

17.   Sterba also states, from time to time, that on an account like Davies's or Murphy's, God would come out as even worse than the greatest villains of history (in virtue of having permitted all the horrendous morally evil consequences of human agents). Even on the assumption that Sterba's response to skeptical theism (see his chp. 5) is sufficient to establish that we have no reason to doubt that the evils we see that apparently violate Moral Evil Prevention Requirements I-III actually violate those requirements (along with qualifications regarding whatever claims about the entailments of value that might be implied by that claim), I think the comparison between God and history's greatest villains is an oversimplification. On the account of God given above, God has violated no moral norms whatsoever. In other words, there is a clear sense in which God has acted in perfect accordance with the requirements of reason pertaining to him. We cannot say the same for history's greatest villains. They *should not* have acted as they did, and in so acting, they flouted those moral norms which were binding on them. Sterba might think this difference irrelevant, but I imagine Davies and Murphy would disagree.

18    Adams (2000, p. 4) suggests that divine *love* is more to the point than moral goodness.

19    What I say here is entirely consistent with claiming, further, that there are multiple reasons for which someone might come to love another. For instance, perhaps the reason I love my spouse was grounded in the fact that she stood out to me, in comparison to others, at some time in the past. This fact, however, is about the *origins* of a relationship of love rather than about the *nature* of the reasons entering such a relationship confers upon oneself. It is the latter topic that is relevant here; that is, whatever the reasons are for why God chose to love humanity, they need not be reasons of love themselves. That is, the explanation of the origin of a relationship of love could very well involve a comparison of some sort, even if God's reasons for action that exist as a result of loving human agents are not themselves comparative.

20    One reviewer notes that Sterba and philosophers such as Murphy and Davies agree that premise (1) of my earlier restatement of Sterba's argument is false. They disagree, however, whether premise (1)'s falsity entails the non-existence of God. This is because Murphy and Davies deny that the concept of God includes moral perfection while Sterba affirms that it does. My contention here is that switching to love is a concession Sterba can afford to make in order to refocus the discussion on the existential premise—i.e., premise (5). Let a '*' by each premise indicate that they have been modified to deal with perfect love rather than moral perfection. Given that (1*)–(5*) would remain an inconsistent set, then theists are forced to either deny that God is perfectly loving—such that God does *not* have reasons of love to promote the welfare of humans, which amount to denying that God loves human agents—or they have to reject one of the other premises. Far fewer theists will be willing to reject (1*) as a misrepresentation of the concept of God, and so, premise (5*) will again be the main point of contention (though, I should note, that even then, premise (3*) may also be denied by some theists (cf. Murphy 2021, chp. 6) for independent reasons of holiness that I cannot go into for the purposes of this paper).

21    This is the fundamental argument in Murphy for reducing the love framework to a special instance of the morality framework.

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
