# Peer review of "Divine Morality or Divine Love? On Sterba’s New Logical Problem of Evil"

_religions, doi:10.3390/rel14020157_

Round 1

Reviewer 1 Report

“Divine Morality or Divine Love? On Sterba’s New Logical Problem of Evil” is well-written, mostly clear, and makes a number of insightful points. However, it’s underdeveloped where development is most needed.

First, following Mark Murphy, the author claims that there are good reasons to suppose that God isn’t a moral agent, which thereby circumvents Sterba’s new logical problem of evil. But this isn’t a great comfort to theists who think that God loves humans, says the author, for such a conception of God only strengthens Sterba’s argument from evil. Yet the author doesn’t treat, at any length, Murphy’s own arguments against the idea that God has requiring reasons of love. All we get is a brief response to one of Murphy’s arguments for the conclusion that God doesn’t have requiring reasons of love from Murphy’s God’s Own Ethics. The author says nothing of Murphy’s treatment of the issue in Divine Holiness and Divine Action, despite the fact that Murphy’s arguments found there are deeply relevant to the author’s argument. If it should turn out that we don’t have sufficient reason to maintain that God has requiring reasons of love, then the author’s argument fails (as the author seems to admit in effect on pp. 5 and 8, e.g.).  

Second, the author doesn’t explain in sufficient detail how God’s possession of non-comparative love supports Sterba’s new logical problem of evil (or the principles that underwrite it). This needs to be corrected. Plus, the author needs to explain in greater detail what it means for God’s love to be non-comparative. Does such a view entail, for instance, that God’s love is entirely indifferent to human or creaturely value (along the same lines often attributed to Anders Nygren), such that God could choose to love platypuses more than God loves humans? Or is the non-comparative view at issue closer to the idea that God’s love is non-comparative or indifferent to human or creaturely value only in certain respects? Might God’s love, e.g., be responsive to creaturely value such that God necessarily loves humans more than platypuses but be such that it is indifferent to value disparities among humans? Arguably, what one decides on such issues will have an impact on the author’s argument.   

Finally, and less significantly, the exchange between Sterba and Davies is not explained with sufficient clarity. From what the author writes, it’s not exactly clear where it is that Sterba and Davies disagree. But, again, this issue is less significant, since it doesn’t bear upon the author’s central argument (like the other two concerns raised). 

Author Response

The Reviewer says,

"First, following Mark Murphy, the author claims that there are good reasons to suppose that God isn’t a moral agent, which thereby circumvents Sterba’s new logical problem of evil. But this isn’t a great comfort to theists who think that God loves humans, says the author, for such a conception of God only strengthens Sterba’s argument from evil. Yet the author doesn’t treat, at any length, Murphy’s own arguments against the idea that God has requiring reasons of love. All we get is a brief response to one of Murphy’s arguments for the conclusion that God doesn’t have requiring reasons of love from Murphy’s God’s Own Ethics. The author says nothing of Murphy’s treatment of the issue in Divine Holiness and Divine Action, despite the fact that Murphy’s arguments found there are deeply relevant to the author’s argument. If it should turn out that we don’t have sufficient reason to maintain that God has requiring reasons of love, then the author’s argument fails (as the author seems to admit in effect on pp. 5 and 8, e.g.)."

Murphy himself states explicitly in Divine Holiness and Divine Action that his argument against the love framework as providing God’s framework of rationality is more fully developed in the earlier book, God’s Own Ethics (e.g., see page 86 of Divine Holiness where he says this about the morality framework—relevant comments also found on page 96; for the same claim regarding the Love Framework, see page 101 and 107). I do not deal with the discussion in the more recent book, despite knowing it very well, because it adds nothing of significance to the present discussion and lacks important detail in the arguments against the love/morality frameworks found in the earlier book.

The Reviewer then says,

"Second, the author doesn’t explain in sufficient detail how God’s possession of non-comparative love supports Sterba’s new logical problem of evil (or the principles that underwrite it). This needs to be corrected. Plus, the author needs to explain in greater detail what it means for God’s love to be non-comparative. Does such a view entail, for instance, that God’s love is entirely indifferent to human or creaturely value (along the same lines often attributed to Anders Nygren), such that God could choose to love platypuses more than God loves humans? Or is the non-comparative view at issue closer to the idea that God’s love is non-comparative or indifferent to human or creaturely value only in certain respects? Might God’s love, e.g., be responsive to creaturely value such that God necessarily loves humans more than platypuses but be such that it is indifferent to value disparities among humans? Arguably, what one decides on such issues will have an impact on the author’s argument.   

I tried to clarify what the argument is doing here, and I claim the stronger thesis: namely, that reasons of love are grounded in comparisons of value of any of the options above (e.g., creatures, intrahuman value, etc.). Perhaps someone thinks that there is something imperfect about loving all things without allowing that one must love some things more than others lest they treat the world in a way that fails to reflect its value structure. I think this is misguided, however. If God desires the good for all creatures, then his love might express itself in different ways according to the natures of the creatures in question (e.g., human beings seem capable of receiving greater amounts and types of goods from God than, say, an earthworm). This, however, strikes me as differences in the beloved—differences in their natural capacities—rather than differences in God’s love of them.

Finally, and less significantly, the exchange between Sterba and Davies is not explained with sufficient clarity. From what the author writes, it’s not exactly clear where it is that Sterba and Davies disagree. But, again, this issue is less significant, since it doesn’t bear upon the author’s central argument (like the other two concerns raised).

Since I am not focused on Sterba's particular disagreement with Davies, I ignore the specifics in the paper. The assumption that moral requirements apply to all rational agents (as opposed to all human agents or whathaveyou) is the only point in that disagreement that is relevant to my paper. This is why I largely leave Davies to the side.

Author Response

The paper is well-written, and its main point valid. Nevertheless I have a few substantive questions:

p. 2: Unclear what the author means by “keeping promises grounds requiring reasons of morality.” If this was not a slip of the pen, then it is a massive claim that needs justification. This should be either amended or left out.

This is an example for understanding the relationship between requiring reasons and justifying reasons. If someone does not share my view that promises ground (prima facie) requiring reasons of morality, they can sub in their preferable example. I’ve added a footnote to that effect.

p. 4: The author distinguishes between requiring and justifying reasons and spends a good bit of time on this distinction. Nevertheless, the author does not make sufficiently clear how exactly this discussion advances their main claim. Presumably, the idea is that authors such as Murphy argue that while God cannot be bound by requiring reasons, God can have justifying reasons regarding God’s love for us. This needs to be brought out more prominently towards the last part of the paper and its connection with the author’s main point made more clearly and prominently.

p. 4: I found the discussion on p. 4, where the author argues one’s justifying reasons can make it rational to act against one’s requiring reasons, questionable. First, An action may be “intelligible” or “understandable,” without that action being rational. For instance, why an agent acts against her own best interests, or against morality in order to get something she deeply wants may be intelligible (we understand she is acting on some very deep-rooted desires, and so we understand why she is doing x) without this action being rational in either a) the sense of it being prudent, or b) the sense of it being moral (if we equate moral action with rational action). Putin’s actions, for instance, are certainly intelligible once we factor in his desire for glory without those actions being rational. The author slides too quickly from intelligibility to rationality. How this paragraph works with the rest of the paper is unclear.

The reviewer has in mind a case where someone has a reason to act one way and makes a mistake about the force of their reasons. Take the case of Putin and suppose his goal is to maximize his glory. It looks like his actions more recently are irrational either because they fail to actually maximize his glory (e.g., maybe to be glorious requires being morally upright in one’s decisions) or because having self-glorification as an end is itself irrational. If Putin’s actions fail to maximize his glory, then Putin has made a mistake about what actions his reasons support (and is consequently irrational). If it is irrational to aim at glory, then even though Putin’s aiming at glory gives him reasons to act in some ways and not others, the practical rationality of his chosen actions (assuming they get him the sort of glory he wants) do not make his chosen actions rational on the whole.

My worry about the reviewers point here, though, is that they have changed the example. The case of promises and Bruce Springsteen tickets can be understood as rational in either sense if read charitably. Going to a Bruce Springsteen concert is not an immoral or intrinsically irrational pursuit. Putin’s aims might be intrinsically irrational, but the Springsteen concert clearly is not. What I claim, then, is that one has practical Springsteen-grounded justifying reasons to miss lunch. I then add that one has weak requiring reasons of morality to keep one’s promise to one’s colleagues. Unless someone thinks that it is never rational to break promises in light of practical considerations (and I think this is clearly too strong a claim to be defensible), then the example works as is for the purposes of illustration.

p. 7: “… my love for my spouse has nothing to do with how much more suitable she is as a spouse for me…” This makes it sound like just anyone else will do. Obviously, if he chose her, she is more suitable than someone else, someone with different beliefs, different values, and furthermore, someone whose company he just does not enjoy. There is obviously some reason or ground of why he is with her rather than someone else. She is unique, no one else makes him feel the way she does, whatever. The problem with comparisons is that they begin from the presupposition that two agents are transposable, or equal, and then you try to show how they are not. What’s problematic with comparisons is that they begin from a starting point of transposability, while when you really love someone you love them as an individual that cannot in any way be replaced or exchanged with another. Also, it is unclear how the author’s discussion here illuminates the idea of divine love.

Footnote 24 has been added to clarify a distinction between the reasons to begin loving someone and the reasons that one has in virtue of loving someone. The former need not be reasons of love, and in virtue of this, they may in fact be comparative. This fact, however, does not affect what I say about reasons of love in the rest of the paper.

Reviewer 3 Report

In his book “Is a Good God Logically Possible?”, Sterba has developed several arguments to the conclusion that “it is not the case that there is an all-good, all-powerful God.” In the first part of his/her paper, the author argues against Sterba’s argument. According to the author “Sterba articulates several moral principles that, on the assumption that God is morally perfect, seem to entail God’s non-existence.” That is, according to the author Sterba assumes that God is a moral agent. In the first part of his/her paper, the author challenges this assumption. In the second part of the paper, the author suggests “an alternative way to frame Sterba’s arguments in terms of divine love.” In the author’s view, this “results in a stronger form of the logical argument.”

Although the paper is well written and suggests a novel response to Sterba’s argument, there are problems with the argument of the paper that, in my opinion, warrant a rejection of the paper.

First of all, the author does not discuss one of Sterba’s formulations of the main idea of his argument, but instead prefers to discuss his/her “own restatement” of the main idea of Sterba’s argument which runs as follows:

“1. An omnipotent and morally perfect being exists (assumed for reductio)

2. An omnipotent being could always act in accordance with Moral Evil Prevention Requirements I, II and III

3. A morally perfect being would always act in accordance with Moral Evil Prevention Requirements I, II and III

4. There exist significant and especially horrendous evil consequences of immoral actions that easily could have been prevented by an omnipotent being without violating Moral Evil Prevention Requirements I, II or III

Therefore,

There is no omnipotent morally perfect being.”

In and of itself it is, of course, in no way problematic to offer an “own restatement” of Sterba’s argument. However, while Sterba’s own formulation of the main idea of his argument (see pp. 189 – 190 of his book) is obviously logically valid, I have doubts about the logical validity of the author’s “own restatement”. Compare, for example, the following argument which has a similar logical structure (i. e. a similar logical form) as the author’s “own restatement” and which is obviously not logically valid:

(1) An omnipotent and morally perfect being exists (assumed for reductio)

(2) An omnipotent being could always act in accordance with the principle that it is forbidden to torture innocent babies

(3) A morally perfect being would always act in accordance with the principle that it is forbidden to torture innocent babies

(4) There exists much happiness in the world that easily could have been prevented by an omnipotent being without violating the principle that it is forbidden to torture innocent babies

Therefore,

There is no omnipotent morally perfect being.

This argument has a similar logical structure (i. e. a similar logical form) as the author’s “own restatement” and is obviously not logically valid. It is puzzling, therefore, that the author claims that his/her “own restatement” is “clearly classically valid”. In my view, the author is well advised to stick to Sterba’s own formulation (which is obviously logically valid).

A second problem with the author’s argument is this: The author seeks to argue that God is not morally perfect because God is no moral agent. This, however, is puzzling because in a way Sterba also argues that God is not morally perfect. In fact, the way the author has presented Sterba’s argument, the assumption that there is an omnipotent and morally perfect being is not a premise but an assumption for reductio. That is, Sterba does not assume that God is morally perfect. Instead, Sterba argues that there is no morally perfect being that is also omnipotent. To put it differently: assuming that there is a God and that God is omnipotent, Sterba argues that God is not morally perfect. It is, therefore, not clear to me which of the premises of Sterba’s argument the author is aiming at. In fact, the author writes that “if Sterba would replace ‘morally perfect’ with ‘perfectly loving’ throughout all the premises of his argument, the question of the argument’s success would no longer turn on premise (1) […]”. That is (given that Sterba’s argument, or, at any rate, the author’s “own restatement” of the argument, does in fact make use of the term ‘morally perfect’), the author suggests that “the question of the argument’s success” turns on premise (1). This is confusing. The author seems to forget that he/she himself has introduced line (1) as an assumption for reductio (not as a premise) and that it does, therefore, not make sense to claim that the argument’s success turns on line (1).

To be sure: My claim is not that it is impossible to improve upon the argument of the paper and to develop an interesting objection against Sterba’s argument. My claim is only that because of the way the argument of the paper is presented it is difficult to say whether it is possible to improve upon the argument of the paper and to develop an interesting objection against Sterba’s argument. Therefore, I recommend rejection of the paper.

Author Response

See the attached document response that includes a slightly altered restatement of Sterba's argument with a proof of its validity in a quantified (minimally) modal logic.

Round 2

Reviewer 3 Report

The author has adequately addressed my concerns about the original version of the paper.

My only recommendation for the author is to stress much more (if possible, already at the beginning of the paper) that - although the revised version of Sterba's argument (in terms of divine love) is better than Sterba's original argument insofar as it avoids the "no-moral-agent-objection" - this is not to say that the revised version of Sterba's argument (in terms of divine love) is a good argument.

This strikes me as very important given that in his/her paper the author has focussed on the "no-moral-agent-objection" and has set aside many other objections that have been formulated against Sterba's argument (and that might equally well be raised against the revised version of Sterba's argument).

In this respect, the author should consider not claiming anymore that the revised version of Sterba's argument is "an even more powerful version of the logical argument" (p. 10; own emphasis). For this seems to suggest that Sterba's argument is a powerful version of the logical argument. The author, however, has not shown that Sterba's argument is a powerful version of the logical argument. The author has only shown that there is a revised version of Sterba's argument that does not fall prey to the "no-moral-agent-objection". In fact, the author adds: "I hereby refrain from issuing any verdicts on the ultimate success of the revised argument" (p. 10). My recommendation is to stress this a little bit more at the beginning of the paper (so that it becomes clearer what the goal of the paper is).

Author Response

Thank you for the helpful comments here. I have added the following lengthy parenthetical to the main text to emphasize that I'm setting aside additional possible objections to the revised argument from Sterba: "(although I do not here suggest that the revised argument ultimately succeeds, for other objections to Sterba’s argument, which I set aside for the purposes of this paper, are relevant in evaluating the revised version of his argument)."

I've also removed the word 'even' from the closing paragraph that you identify as carrying the problematic implicature. Since it doesn't follow from an argument's being more powerful than another that either of them is powerful, and given the qualifications to the effect that I'm skeptical of this at the beginning of the paper, I have chosen to leave it open whether I think that Sterba's argument is powerful. It's true that I haven't argued that it is, but I also don't want to give the impression that it is not powerful, as I haven't argued for that conclusion either.